# Elevating the Levels of Calcium Ions Exacerbate Alzheimer’s Disease via Inducing the Production and Aggregation of β-Amyloid Protein and Phosphorylated Tau

**DOI:** 10.3390/ijms22115900

**Published:** 2021-05-31

**Authors:** Pei-Pei Guan, Long-Long Cao, Pu Wang

**Affiliations:** College of Life and Health Sciences, Northeastern University, Shenyang 110819, China; guanpp@mail.neu.edu.cn (P.-P.G.); llcao_bio@163.com (L.-L.C.)

**Keywords:** calcium ions, transporters, mechanisms, Alzheimer’s disease, review

## Abstract

Alzheimer’s disease (AD) is a neurodegenerative disease with a high incidence rate. The main pathological features of AD are β-amyloid plaques (APs), which are formed by β-amyloid protein (Aβ) deposition, and neurofibrillary tangles (NFTs), which are formed by the excessive phosphorylation of the tau protein. Although a series of studies have shown that the accumulation of metal ions, including calcium ions (Ca^2+^), can promote the formation of APs and NFTs, there is no systematic review of the mechanisms by which Ca^2+^ affects the development and progression of AD. In view of this, the current review summarizes the mechanisms by which Ca^2+^ is transported into and out of cells and organelles, such as the cell, endoplasmic reticulum, mitochondrial and lysosomal membranes to affect the balance of intracellular Ca^2+^ levels. In addition, dyshomeostasis of Ca^2+^ plays an important role in modulating the pathogenesis of AD by influencing the production and aggregation of Aβ peptides and tau protein phosphorylation and the ways that disrupting the metabolic balance of Ca^2+^ can affect the learning ability and memory of people with AD. In addition, the effects of these mechanisms on the synaptic plasticity are also discussed. Finally, the molecular network through which Ca^2+^ regulates the pathogenesis of AD is introduced, providing a theoretical basis for improving the clinical treatment of AD.

## 1. Introduction

Alzheimer’s disease (AD), commonly known as dementia, is a neurodegenerative disease with a high incidence rate. AD may share common biological pathways and is often associated with diabetes and other comorbidities [1] Clinically, cognitive dysfunction is the main feature [2]. Although the pathogenesis of AD has not been definitely determined, it is generally believed that the pathogenesis of AD is related to the excessive production and deposition of β-amyloid protein (Aβ) and hyperphosphorylated tau protein [3]. On the one hand, Aβ is produced mainly through the amyloid metabolic pathway when the amyloid precursor protein (APP) is cleaved by β-secretase and γ-secretase to produce Aβ monomers [4]. On the other hand, the tau protein is hyperphosphorylated through the action of cyclin-dependent kinase 5 (Cdk5) and glycogen synthase kinase (GSK) 3β [5]. Both the Aβ and phosphorylated tau proteins have the ability to self-aggregate. Through this self-aggregation, they gradually form oligomers and fibers, which are deposited as β-amyloid plaques (APs) and neurofibrillary tangles (NFTs), respectively [6]. The formation of oligomers and fibers can mediate the pathological progress of AD by affecting the function of glial cells and neurons [7].

A series of studies have shown that the onset of AD is related to aging; an unhealthy lifestyle, including smoking and drinking; health status, such as degree of heart disease, hypertension, obesity and diabetes; and genetic factors, such as APOE4 expression [8,9,10,11]. For the production of Aβ, mutations in APP and presenilin (PS), including PS1 and PS2, are the decisive factors [12,13,14]. However, the phosphorylation of tau protein greatly affects the stability of microtubes in neurons, resulting in neuronal tangles [15]. In addition to the production and deposition of Aβ and phosphorylated tau protein, many metal ions contribute to metabolic disorders [16]. In PS-mutant AD brain tissue, a Ca^2+^ metabolic disorder was evident before the formation of APs or NFTs [17], This observation was further corroborated by a series of evidence in different AD animal models [18,19,20], which indicated that the metabolic disorder caused by Ca^2+^ located in the cytoplasm might be the cause of AD. Based on this hypothesis, previous studies have shown that Ca^2+^ influx can increase the production and aggregation of Aβ and phosphorylated tau protein and thus affect the learning and memory of patients with AD [17,21,22].

Moreover, the imbalance of Ca^2+^ leads to dysregulated metabolism that affects many neurophysiological functions related to AD, including the regulation of neuroinflammation, response to neuronal injury, neuronal regeneration, neurotoxicity, autophagy and synaptic plasticity [23,24,25,26,27]. The multifunctional AD-related neuropathological function of Ca^2+^ may be directly or indirectly mediated by Aβ and/or phosphorylated tau proteins. As the main pathological features of AD, monomeric or aggregate Aβ and phosphorylated tau proteins show regulatory effects on neuroinflammation, neuronal injury, neuronal regeneration, neurotoxicity, neuroprotection, autophagy and neural plasticity [16]. Either directly or indirectly, Ca^2+^ is involved in the regulation of these neuropathological functions through its specific transporters. Therefore, this review mainly explores the molecular mechanisms by which a Ca^2+^ imbalance in AD affects the regulation of Aβ, tau, and neural plasticity, specifically from the perspective of Ca^2+^ transporters in cell, mitochondrial, endoplasmic reticulum (ER) and lysosomal membranes.

## 2. APP Metabolic Products Including Aβ Facilitated the Influx of Ca^2+^ into the Neurons of AD Animals and Patients

The concentration of Ca^2+^ is strictly regulated under physiological conditions, whereas Ca^2+^ concentration is obviously elevated in the brains of AD patients and APP/PS1 Tg mice [19]. Kuchibhotla et al. found that Ca^2+^ is significantly increased in the dendrites and dendritic spines of neurons of APP/PS1 Tg mice [28]. In view of their observation, the natural question that arises is: What is the reason for Ca^2+^ elevation during the course of AD development and progression? It has been reported that Aβ_1–40_ has the ability to upregulate the influx of Ca^2+^ in rat cortical synaptosomes and cultured cortical neurons [29,30]. Moreover, the Aβ_25–35_ peptide has an effect similar to that of Aβ_1–40_, which can promote Ca^2+^ influx by activating L- and T-type Ca^2+^ channels in rat hippocampal slices [31]. Similar to the results in vivo, Aβ increased the Ca^2+^ influx in PC12 and SH-SY5Y cells in vitro [32,33]. In addition to activating ion channels, Aβ has the ability to activate PKA, which increases Ca^2+^ influx through L-VGCCs by activating calcium-binding proteins [34].

Because of the self-aggregating characteristics of Aβ, the concentration of Ca2+ in the spines and dendrites of cortical pyramidal neurons around APs is higher than the normal value in adjacent resting neurons [22]. In addition to the effect of APs on Ca^2+^ in neurons, Bacskai and his colleagues quantitatively measured the resting-state Ca^2+^ concentration in astrocytes of APP/PS1 mice and observed the overall response of astrocytes to AP deposition. The results showed that the concentration of Ca^2+^ in the astrocytes of 6-month-old mutant mice was elevated compared to that of the WT controls [35]. It was confirmed that the resting level of Ca^2+^ reached 247 nmol/L in the cortical neurons of 3×Tg mice, which is twice that of the cortical neurons of non-Tg controls (110 nmol/L) [22]. Taking advantage of live cell imaging, the level of Ca^2+^ was found to be elevated in neurites, which were 20 μm from the central AP region, indicating the critical roles of APs in the homeostasis of Ca^2+^ in the spines and dendrites of neurons [36]. In astrocytes of 6-month-old APP/PS1 mice, Ca^2+^ was elevated in response to the deposition of APs [35]. In transient occlusion of the middle cerebral artery (MCAO) of hAPP695 transgenic (Tg) rats, Ca^2+^ colocalized with APs and was deposited in the thalamus [37]. Arispe et al. found that the aggregates of Aβ_1–40_ and Aβ_1–42_ can form a cation channel on the surface of an artificial lipid membrane that allows the passage of Ca^2+^ [38]. However, the channel showed low selectivity, and thus it also permitted the passage of Li^+^, K^+^ and Na^+^ [39]. In SH-SY5Y cells, oligomeric Aβ cannot selectively increase the Ca^2+^ permeability of cellular membranes, thereby increasing both Ca^2+^ influx from the extracellular space and Ca^2+^ leakage from intracellular Ca^2+^ stores [35]. The pore formation of Aβ was confirmed and corroborated by atomic force microscopy [40], electron microscopy [41,42] and a theoretical model [43,44]. For example, high-resolution transmission electron microscopy revealed the presence of Aβ pores distributed in situ in the cell membranes of post-mortem AD patients [36]. In addition, the formation of Aβ pores is also considered a mechanism of neurotoxicity induction, which destroys cell homeostasis by inducing the leakage of Na^+^, K^+^ and Ca^2+^ through this highly conductive channel [45]. This observation reinforces the extreme toxicity of Aβ oligomers, which potentially disrupts the homeostasis of Ca^2+^ in neurons [46,47,48]. The formation of Aβ pores is enhanced by the presence of phosphatidylserine, a cell surface marker of early apoptosis [49]. However, this kind of pore can be blocked by Zn^2+^, because Zn^2+^ can form a complex with Aβ to prevent the aggregation of Aβ, which inhibits the insertion of Aβ oligomers into the membrane, leading to the formation of pores [50,51,52,53]. In addition, the extent of the pore-forming activity of Aβ in the lipid bilayer is inversely proportional to the cholesterol level in the lipid mixture. Treatment with cyclodextrin significantly enhanced the toxicity of Aβ in PC12 cells by decreasing or inhibiting the increase in the cholesterol level of these cells [54]. In contrast, Kawahara and Kuroda found that increasing the cholesterol content on the surface of the cell membrane significantly reduced Aβ-induced Ca^2+^ influx [55].

In addition to Aβ, sAPP is involved in regulating the homeostasis of Ca^2+^. For instance, sAPP mediates the effects of glutamate on the regulation of the homeostasis of Ca^2+^ by increasing the production of cyclic (c) GMP to activate K^+^ channels, which results in reduced Ca^2+^ levels in hippocampal neurons [56]. In addition, it has been reported that a PS1 mutation is a key factor for sAPP stabilization of the homeostasis of Ca^2+^ in hippocampal neurons [57]. A possible explanation for this effect may involve the reversed regulation of APP695 and InsP3R genes at the mRNA and protein levels during differentiation [58]. The APP intracellular domain (AICD), which is released after InsP3R cleavage of APP may act as a transcription factor to activate the Ca^2+^ signaling system [59,60]. As the cleavage fragments of APP are produced by different secretases, PSEN2 mutation has shown its effects on impairing the fusion between autophagosomes and lysosomes in PSEN2^T122R^ mutated SH-SY5Y cells [61]. However, these effects are not caused by the activity of g-secretase but by decreasing the Ca^2+^ released from ER in an ER-dependent mechanism [61].

## 3. Ca^2+^ Transporters on the Surface of the Nerve Cell Membrane Are Responsible for Promoting the Influx of Ca^2+^ during the Course of AD Development and Progression

In addition, there are many natural Ca^2+^ transporters on the surface of the nerve cell membrane (Figure 1). As an antagonist of N-methyl-D-aspartic acid receptor (NMDAR), memantine significantly inhibits Ca^2+^ influx and was the first Food and Drug Administration (FDA)-approved drug for the treatment of moderate to severe AD in patients [62]. This drug was designed because Aβ can interact with endogenous Ca^2+^ channels in the cell membrane to increase NMDAR-dependent Ca^2+^ influx [63]. On the basis of this drug, memantine nitrate-06 (MN-06) was developed to protect the neurotoxicity against glutamate via inhibiting the influx of Ca^2+^ and decreasing the activity of PI3-K/Akt/GSK-3β pathways in primary cultured rat cerebellar granule and hippocampal neurons [64]. Although Aβ oligomers can promote Ca^2+^ influx through NMDAR channels in a short period of time [65], sustained exposure to Aβ oligomers decreases the expression of NMDAR, the extent of Ca^2+^ influx and the glutamate current in neurons [66,67,68]. In addition to targeting NMDARs, the antagonists of amino-3-hydroxy-methylisoxazole-4-propionate receptor (AMPAR), such as LY451395, LY450108 and S18986, reverse Ca^2+^ influx in AD animal models [69,70,71,72].

In addition to glutamate receptors, there are a series of voltage gated Ca^2+^ channels (VGCCs) on the surface of the cell membrane that mediate the transportation of Ca^2+^. For example, Aβ blocked presynaptic P/Q-VGCC, which resulted in reduced Ca^2+^ influx into hippocampal neurons [73]. In contrast, Aβ_1–40_ concurrently enhanced the high threshold and low conductance of N- and T-VGCC and the high conductance of L-VGCC, which resulted in an increasing postsynaptic Ca^2+^ response in cortical neurons [29,74,75]. In addition, Aβ impaired ion motive ATPases, which resulted in membrane depolarization and the opening of NMDAR pores and VGCCs, leading to an influx of Ca^2+^ and impaired Ca^2+^-ATPase, which resulted in inhibited Ca^2+^ efflux in primary cultured neurons and synaptosomes of an adult post-mortem hippocampus [76]. Although the mechanism by which CALHM1 serves as a cation channel in the brain is not completely clear, it has been reported as a pore-forming subunit whose activation can regulate Ca^2+^ influx, and it is regulated by the voltage and extracellular Ca^2+^ concentration of mouse cortical neurons [77]. As a potential Ca^2+^ transporter, it is further confirmed in CALHM1 knocking out mice [78]. As an important biomarker of AD, APOE does not directly regulate Ca^2+^ influx as a canonical cation channel, but it can promote the influx of Ca^2+^ by activating P/Q-VGCC in neurons [79,80]. In primary cultured astrocytes of APOE4^−/−^ mice, APOE4 was found to be responsible for impairing neurons after brain injury [81].

## 4. ER Is an Important Reservoir to Elevate the Levels of Ca^2+^ in the Neurons of AD

As an important reservoir of Ca^2+^ in neurons, endoplasmic Ca^2+^ can pass through InsP3Rs and ryanodine receptors (RyRs) to enter the cytosol (Figure 2). In the resting state, the intracellular level of Ca^2+^ remains at a relatively low level, between 50–300 nM. After activation, Ca^2+^ is mainly stored in the endoplasmic reticulum (ER), where the concentration of Ca^2+^ is approximately 100–500 nM and can be released into the cytoplasm through InsP3R and RyR [82,83]. Previous studies have shown that Aβ_25–35_ induces the transportation of Ca^2+^ in association with the activation of phospholipase C (PLC) and the production of inositol triphosphate (InsP3) [84]. In neurons, the addition of experimental Aβ significantly increased the Ca^2+^ response induced by InsP3R [85]. More specifically, exposing RyRs to Aβ_1–42_ increases the probability of channel opening, which results in an increased Ca^2+^ flux [86]. Similarly, Aβ aggregates have the ability to increase Ca^2+^ flux from the ER via InsP3R and RyR in human brain tissues and cells and in hippocampal CA1 pyramidal neurons [82,83,87,88].

In addition to Aβ, PS1 exhibits the ability to interact with three key components of the Ca^2+^ signaling cascade, namely, InsP3R [89,90], RyR [91,92,93] and sarcoplasmic/endoplasmic reticulum calcium ATPase (SERCA) [94]. Recently, Cheung et al., found that PS can physically interact with InsP3R to stimulate its gating activity, which results in an increase in Ca^2+^ even though there is no increase in Ca^2+^ in the lumens of the ER [90]. In SH-SY5Y cells, a PS1 mutation enhances the activity of PLC, leading to an increase in the level of IP_3_, which results in the release of Ca^2+^ from the ER [95]. Similarly, a PS mutation can stimulate Ca^2+^ release from the ER via InsP3R and RyR [93,96,97]. In the ER membrane, there is a sarcoplasmic/endoplasmic reticulum ATPase (SERCA) pump in addition to InsP3R and RyR. In CHO cells, PS mutants bound to the SERCA pump, which disturbed the balance of Ca^2+^ [94]. In 3 × Tg mice, InsP3R and RyR mediated the release of Ca^2+^ from the ER, from which it entered the cytosol [90,91]. Interestingly, APOE4 may trigger the release of ER-Ca^2+^ via RyR, which promotes the formation of APs and NFTs [98,99,100,101]

However, the depletion of ER Ca^2+^ induces a continuous influx of extracellular Ca^2+^ into the cytoplasm by activating a classical store operated Ca^2+^ entry (SOCE) pathway. This process initially requires the sensor molecule of canonical systemic Ca^2+^ interactions in the ER (stromal interaction molecule, Stim) to sense ER Ca^2+^ depletion, which leads to activated Ca^2+^ channels on the surface of the cell membrane, such as Ca^2+^ release-activated Ca^2+^ (CRAC) channels, also known as calcium channel protein 1 (CRACM1, Orai1) channels [102,103]. Although Stim-related proteins, including Orai and TRPC, are located on the surface of the cell membrane, we prefer to discuss their roles in Ca^2+^ transportation because of their close relationship with the ER. As expected, SOCE disruption by the Stim1^D76A^ mutation attenuated Ca^2+^ entry in primary neurons from AD mice with human mutant-PS1-knock-in skin fibroblasts from familial AD patients [104,105]. Other studies have shown that the expression level of Stim2 was downregulated by this PS1 mutant, which resulted in insufficient signals transmitted to the plasma membrane to activate SOCE, leading to reduced influx of Ca^2+^ when Ca^2+^ was depleted from the ER [106]. Moreover, PS1^ΔE9^ mutation induces the influx of Ca^2+^ via activating Stim1 in a SOCE-dependent mechanism in mouse hippocampal neurons [107]. Although there was no direct evidence showing their association with the activation of SOCE, TRPC3 and TRPC6 played roles in regulating the homeostasis of intracellular Ca^2+^ [108,109,110].

## 5. Mitochondria and Lysosomes Also Act as Important Organelles for Regulating the Dyshomeostasis of Ca^2+^ during the Development and Progression of AD

In addition to the ER, mitochondria and lysosomes play important roles in the regulation of Ca^2+^ homeostasis, which has been reviewed in detail in a previous study [22] (Figure 3 and Figure 4). In brief, there is evidence showing that the PS1^L286V^ mutant can promote disorders in Ca^2+^ homeostasis in neurons by damaging mitochondria [111,112]. In PS1^M146L^ mutant lymphoblasts, activation of InsP3R results in opening mPTP transporters in mitochondria [113]. In a series of AD-related mice and cell models, VDAC and MCU mediated the mitochondrial uptake of Ca^2+^ [114,115,116]; the Na^+^/Ca^2+^ exchanger is critical for Ca^2+^ export across the inner mitochondrial membrane (IMM) [117,118,119]; and the mitochondrial permeability transition pore (mPTP) is critical for the efflux of Ca^2+^ from neuronal mitochondria [120]. Although there is no direct evidence showing the involvement of Aβ in mitochondrial Ca^2+^ transportation, Aβ has the ability to open the mPTP, leading to the release of cytochrome C and caspases from mitochondria [100,121]. This evidence also indicates that the excessive accumulation of Aβ may be involved in the regulation of mitochondrial Ca^2+^ homeostasis. In contrast to that internalized by mitochondria, the Ca^2+^ uptake into lysosomes is mainly realized by the cooperation of a vacuolar type H^+^-ATPase (v-ATPase) and a putative Ca^2+^/H+ exchanger (CAX) [122,123]. The excretion of Ca^2+^ from lysosomes is mainly realized by TRPML and TPC [124]. When Ca^2+^ flows out of lysosomes through these VGCCs, defective autophagic lysosomes form, leading to autophagy [125]. Furthermore, the mutation or deletion of PS1 in AD leads to the disequilibrium of lysosomal Ca^2+^ by reducing the activity of the v-ATPase proton pumps on the lysosome, leading to AD pathogenesis [126]. In PS1 and 2 double knockout neurons, the number of lysosomal Ca^2+^ stores were significantly decreased, which resulted in a damaged autophagy process [127]. The imbalance of these processes (Table 1) affects the clearance of disease-related proteins in the pathogenesis of AD.

## 6. The Roles of Ca^2+^ in the Production and Deposition of Aβ during the Course of AD Development and Progression

An increase in Ca^2+^ levels is functionally related to most pathological features and pathogenic factors of AD, such as presenilin and APP mutations, APOE4 expression, CALHM1 mutation, Aβ plaque formation, tau hyperphosphorylation, apoptosis and synaptic dysfunction [100]. In the following discussion, we discuss these features individually. This section focuses on the regulation of Ca^2+^ metabolism during the production and deposition of Aβ and phosphorylation of tau protein (Table 2). In HEK293 cells overexpressing human APP, the Ca^2+^ ion carrier A23187 can increase Aβ production by increasing intracellular free Ca^2+^ [134,135]. In primary cultured neurons from 3 × Tg mice, Ca^2+^ chelator, BAPTA/AM and TRPV1 antagonist, capsazepine lowered the levels of Aβ and phosphorylated tau [136]. In SH-SY5Y neurons cultured in vitro, increased Ca^2+^ levels also led to an increase in the production of Aβ [36]. Other studies have shown that Ca^2+^ can promote the formation of the Aβ_1–40_ oligomer, which is also the main cause of AD neurotoxicity [137]. In addition, the increase in intracellular Ca^2+^ levels can also trigger the aggregation of Aβ, which forms fibrils, indicating that Ca^2+^ instability is a possible cause of sporadic AD [138]. The results of circular dichroism (CD) spectra demonstrated that 1–2 mM Ca^2+^ have the ability to alter the unfolded Aβ_1–42_ to β-sheet structure, which results in shortening the time of forming Aβ_1–42_ fibrils by thioflavin T staining [139]. During the formation of Aβ fibrils, Ca^2+^ seemed to accelerate the seeding effects of Aβ_1–42_ in AD [139].

## 7. Ca^2+^ Transporters on the Cell Membrane Are Potentially Contributed to the Role of Aβ in the Pathogenesis of AD

Since Ca^2+^ has been shown to play a role in the production and aggregation of Aβ, transporters on the surface of the cell membrane must have the potential to regulate the role of Aβ in the pathogenesis of AD. In SH-SY5Y cells and APP23 Tg mice, memantine, an antagonist of NMDAR, showed an inhibitory effect on the production of Aβ [138,142]. This result confirmed the theory that the activation of NMDAR can induce the production of Aβ [143]. In addition, a recent study with an AD Tg mouse model showed that Ca^2+^-permeable (CP) AMPAR was abnormally expressed in the brains of APP/PS1 Tg mice [164,165]. In line with this finding, recent studies have found that the direct injection of Aβ oligomers into hippocampal neurons in the CA1 region leads to the rapid insertion of CP AMPAR into synapses [164,165]. The activation of AMPAR can increase the α-secretase cleavage of APP, thereby inhibiting the production of Aβ [144]. In addition to these glutamate receptors, the CALMH1^P86L^ polymorphic protein also increased the production of Aβ [138,145]. In rat cortical neurons, L-VGCC promoted Aβ production by increasing the Ca^2+^ influx [138,166] In this scenario, APOE, as a transmembrane protein, also participates in the regulation of Aβ production [34,146].

## 8. Ca^2+^ Leakage from ER Modulates the Production and Deposition of Aβ via Activating Ca^2+^ Transporters on ER

In addition to extracellular Ca^2+^ influx, the ER, as an intracellular reservoir, plays a regulatory role in the production of Aβ. For example, knocking out the expression of InsP3R in Sf9 and DT40 cells significantly reduced Aβ production [90]. In addition, previous studies have shown that RyR protein and mRNA expression levels were significantly increased in SH-SY5Y neuroblastoma cells and Tg2576 mice overexpressing wild-type βAPP or βAPPswe [149]. RyR, another important Ca^2+^ transporter on the surface of the ER membrane, also regulates Aβ production [96]. By inhibiting RyR activity, dantrolene decreased the activity levels of β- and γ-secretases and the formation of APs [148,149]. In AD patients with mild cognitive impairment, RyR2 expression is increased [167,168]. In mutant-APP-overexpressing SH-SY5Y neurons, the post-translational modification of RyR2 can affect Ca^2+^ leakage from the ER, leading to reduced production of Aβ from APP [150]. In addition, it has been reported that enhancing the binding of FKBP12.6 with RyR2 can stabilize the leakage of Ca^2+^ from the RyR2 channel, leading to the formation of fewer APs [150]. In addition to RyR2, the RyR3 level showed an upward trend in the hippocampus of several AD mouse models [96,148,149]. In contrast to RyR2, some studies have shown that knocking out RyR3 reduces the formation of APs in the brains of AD mouse models [151]. By knocking out the expression of RyR3, RyR3 was found to exert a neuroprotective effect in the early stage of AD but promoted the development of AD in the late stage in a 3 × Tg mouse model [151,169]. Thapsigargin inhibition or siRNA knockout of SERCA, a Ca^2+^ channel in the ER, resulted in a decrease in Aβ production, while SERCA overexpression increased Aβ production [94]. Thapsigargin, a compound that inhibits Ca^2+^ uptake into the ER through SERCA, can increase the effects of caffeine on stimulating the release of Aβ by increasing the level of Ca^2+^ in the cytoplasm [135]. These conflicting reports are reconciled by previous reports showing that lower concentrations (10 nM) of thapsigargin stimulated the formation of Aβ, whereas higher concentrations (20 nM) of thapsigargin inhibited the production of Aβ in APP-overexpressing CHO cells [152].

On the basis of SOCE, the overexpression of Stim1 and Orai1 can accelerate the production and deposition of Aβ [105]. In PS1^M146V^-overexpressing hippocampal neurons, SOCE is required for maintaining the morphology of mushroom spines, which results in modulating the production of Aβ and promoting memory functions [153,154]. In human neuroblastoma cells, the influx of Ca^2+^ mediated by SOCE can reduce the secretion of Aβ, suggesting that the loss of SOCE in the pathogenesis of AD leads to the production of Aβ and accelerates the onset of AD [155,156,157]. Consistent with this hypothesis, inhibition of SOCE by overexpressing Orai2 results in the increased production of Aβ_1–42_ in SH-SY5Y and human neuroglioma H4 cells, suggesting a potential way to rescue the defects of AD and prevent the formation of APs by downregulating the expression of Orai2 [157,158]. 

## 9. Ca^2+^ Transporters on the Membranes of Mitochondria Are Also Involved in Regulating the Production and Deposition of Aβ during the Course of AD Development and Progression

In mitochondria, the abnormal interaction of voltage-dependent anion channel 1 (VDAC1) with Aβ and phosphorylated tau has the ability to induce the dysfunction of mitochondria during the course of AD development and progression [170]. In addition, Aβ can induce the opening of mPTP, which results in enhanced permeability of the brain mitochondria [171,172]. These observations indicated that Aβ might induce the efflux of Ca^2+^ from mitochondria, which enhances the pathogenesis of AD. In support of this hypothesis, a report suggested that reduced VDAC1 expression in VDAC1^+/−^ mice decreased the mRNA expression levels of AD-related genes, including βAPP, Tau, PS1, PS2 and BACE1, compared with their expression levels in VDAC1^+/+^ mice [159]. Furthermore, in primary cultured neurons and APP/PS1 Tg mice carrying human APP^KM670/671NL^ and PS1^L166P^ mutants, treatment with dutasteride decreased the formation of APs by disrupting the function of the mPTP [160].

## 10. The Roles of Ca^2+^ in Regulating the Phosphorylation of Tau

Apart from the production and deposition of Aβ, Ca^2+^ also induced the phosphorylation of tau via the GSK3β-activating pathway in SH-SY5Y cells [100,161]. In addition, a similar phenomenon was observed in primary cultured hippocampal neurons and immortalized GnRH neurons (GT1–7 cells) [162]. Similarly, we found that mPGES-1/PGE_2_/EPs/CDK5/p35/p25 signaling cascades mediated the effects of Ca^2+^ in stimulating the phosphorylation of tau in n2a and APP/PS1 Tg mice [19]. Furthermore, Ca^2+^ triggered Ca^2+^-activated kinases, which mediated the phosphorylation of tau, leading to the formation of NFTs in AD mouse models [100,163]. Although there are few reports showing the involvement of transporters in mediating the effects of Ca^2+^ on the phosphorylation of tau, there is evidence suggesting that AMPAR mediates the effects of Ca^2+^ on the phosphorylation of tau in PS1^mut^-knock-in mice [140,141]. Furthermore, alterations to the RyR Ca^2+^ release channel correlate with the formation of NFTs in AD patients [147]. On the basis of these observations, multiple transporters may mediate the effects of Ca^2+^ on the production and deposition of Aβ and hyperphosphorylated tau during the course of AD development and progression.

## 11. Ca^2+^ Accelerates the Cognitive Decline Associated with AD

As Ca^2+^ has been observed to be critical for the production and deposition of Aβ and hyperphosphorylated tau via its transporters, we also address its roles in the learning ability and memory of AD patients and experimental models (Table 3). In aging people, elevated levels of serum Ca^2+^ is thought to be associated with cognitive decline [173,174]. In AD patients, disorders of Ca^2+^ metabolism are also reported to be associated with dementia [175]. For this reason, Aβ oligomers were identified as critical for the influx of Ca^2+^ that results in impaired learning and memory through the inhibition of LTP, a form of synaptic plasticity [176,177,178]. Because of the presence of Aβ, Ca^2+^-dependent enzymes located in the spine, such as calpain, are associated with synaptic dysfunction. Treatment with calpain inhibitors improved learning ability and memory by inducing LTP in Aβ-treated APP/PS1 Tg mice [179]. 

## 12. Transporters on the Cell Membrane Mediated the Effects of Ca^2+^ on Inducing the Cognitive Decline of AD

Since the levels of Ca^2+^ are increased by activating calcineurin (CaN), the effects of Aβ in inducing deficits in learning and memory were blocked by inhibitors of CaN in APP/PS1 Tg mice [180]. Activation of the Ca^2+^-dependent protein phosphatase calcineurin (CaN) potentially impaired the cognition of AD by eliminating both NMDA and AMPA receptors through endocytosis [181]. In addition to CaN, NMDAR-specific antagonists showed beneficial effects on learning ability and memory in rats [182,183]. Consistent with this observation, blocking NMDAR attenuated cognitive decline by restoring the metabolic balance of Ca^2+^ in AD patients and AD mouse models [184,185]. The sustained expression of another glutamate receptor serving as a Ca^2+^ transporter, CP-AMPAR, in the early stage of AD accelerated the onset of neuronal network dysfunction and neuronal excitotoxicity, leading to successive cognitive decline by dysregulating the flux of Ca^2+^ [186]. These observations indicate that glutamate receptors, including NMDAR and AMPAR, are critical for mediating the effects of Ca^2+^ dysregulation on the learning ability of AD patients.

In addition, the increase in L-type Ca^2+^ currents in CA1 synapses leads to a decrease in cognitive function in 3 × Tg AD mice [187]. Furthermore, treatment with nifedipine, a calcium channel blocker, attenuated cognitive impairment in KK-A(y) mice, a type 2 diabetic mouse model [188]. These observations confirmed that nimodipine can enhance the learning ability of mild-to-moderate AD patients [189,208]. Similarly, ST101, an inhibitor of T-VGCC, can attenuate cognitive decline by enhancing LTP and the autophosphorylation of CaMKII in rats [190]. As an inhibitor of NMDAR, MK-801 attenuates cognitive decline by decreasing the concentration of Ca^2+^ in mice with traumatic brain injury (TBI) [191]. In Cav 2.1-knockout mice, ablation of Cav2.1 voltage-gated Ca^2+^ channels enhanced learning ability by reducing intracellular Ca^2+^ levels [192]. SB366791, a specific TRPV1 antagonist, ameliorated the poor cognitive performance of dopamine D3 receptor (D3R)-knockout mice [193]. Although it is not regarded as a canonical Ca^2+^ transporter, APOE4 shows the ability to worsen cognitive function by increasing serum Ca^2+^ levels in older people [194]. Moreover, the CALHM1^P86L^ polymorphic protein has been found to be associated with AD in the ethnic Chinese Han population, even though no direct evidence has shown a relationship between Ca^2+^ and learning ability [195].

## 13. Ca^2+^ Transporters on ER Are also Involved in Impairing the Memory of AD

For intracellular stores, the generation of InsP3 can enhance memory loss by activating the release of intracellular Ca^2+^ through a metabotropic glutamate receptor-activating mechanism [196]. As the natural ligand of InsP3R, InsP3 usually exerts its effects via its receptor to impair memory by triggering the release of Ca^2+^ from the ER in AD patients [197]. In addition to InsP3R, RyR was also shown to be critical for the cognitive decline of AD patients and mouse models [209]. For example, the expression of RyR2 was upregulated in patients with mild cognitive impairment and AD [167]. In addition, an inhibitor of RyRs, dantrolene, enhanced the learning ability of an AD mouse model via the rescue of lost synaptic plasticity [198]. To clarify the effect, the expression of RyR3 was knocked down, which resulted in impaired social behavior and memory in rats [199]. This result seemed to conflict with the outcomes induced by treatment of RyRs inhibitors. However, these conflicting results are reconciled by the fact that RyR3 knockdown induces the mRNA expression of RyR2 in the hippocampus of rats completing water maze tests compared with the swimming rat controls [200]. These observations demonstrate the key roles of RyR2 in affecting the learning ability of organisms affected by AD. In addition to its expression, the post-translational modification of RyR can induce cognitive deficits by stabilizing the leakage of Ca^2+^ from the ER [150]. Ca^2+^ depletion by InsP3R and RyR stimulates SOCE. Accordingly, the reduced expression of synaptic STIM2 and impaired SOCE destabilized mushroom spines, which resulted in reduced LTP-mediated memory formation in PS^mut^ mice [106,107,201]. Consistent with this observation, attenuation of SOCE in AD neurons might account for the cognitive decline associated with AD, suggesting possible roles for SOCE in regulating memory functions [202].

## 14. Ca^2+^ Transporters on Mitochondria and Lysosomes Potentially Contribute to the Memory Loss of AD

Although there is no direct evidence to show the relationship between Ca^2+^ from mitochondria and lysosomes and the learning ability of AD patients, to the best of our knowledge, VDAC1 is a hub protein that interacts with more than 150 other proteins, including phosphorylated tau, Aβ, and γ-secretase, and it contributes to their toxic effects, triggering cell death and potentially leading to the dementia characteristic of AD [203]. In addition, DS16570511 and DS44170716 inhibit Ca^2+^ uptake in mitochondria by MCU, which resulted in the inhibition of Ca^2+^-induced mPTP opening and rescued cells from apoptotic death [204]. For lysosomes, tetrandrine and NED-19 inhibited TPCE2 to re-acidify the lysosome environment and reverse dysregulated autophagy [206], which is important for the degradation of aggregated proteins during the course of AD development and progression [207,210]. On the basis of these observations, Ca^2+^ has the ability to modulate the learning ability of AD patients via the functions of its transporters.

## 15. The Roles of Ca^2+^ in Synaptic Plasticity

In neuroscience, synaptic plasticity refers to the connection between nerve cells, whose strength can be adjusted by cell-adhesion molecules, cytoskeletal proteins, ion channels and various receptor proteins [211,212]. Indeed, emerging evidence has revealed the central roles of Ca^2+^ in mediating the synaptic dysfunction in AD [213]. Given the roles of Ca^2+^ in producing Aβ, mutations of APP and PS1 have shown led to disruptions of synaptic processes by controlling the homeostasis of Ca^2+^ during the course of AD development and progression [214]. In addition, the C-terminus of APP has the ability to impair LTP in mice [215]. In fact, Aβ induces Ca^2+^ influx, which results in activating LTD, leading to erased memories in the early cognitive decline of AD patients [28]. Similarly, Aβ oligomers mediate the inhibitory effects of Ca^2+^ on LTP in hippocampal slices [176]. By knocking out the expression of PS1 in mice, LTP is reduced because of the disrupted function of the ER, Ca^2+^ leakage and reduction in the ER Ca^2+^ pool in AD [216]. In contrast, BAPTA-AM, as a chelator of Ca^2+,^ induced LTP in aged rat hippocampal slices [217]. All this evidence demonstrated the effects of Ca^2+^ on synaptic plasticity.

## 16. Ca^2+^ Transporters on the Cell Membrane Are Involved in Regulating the Synaptic Plasticity

CaN is a member of the serine/threonine protein phosphatase family. It is a unique serine/threonine protein phosphatase that is regulated by Ca^2+^ and calmodulin. Currently, it is a multifunctional signaling enzyme, especially in regulating synaptic plasticity. For example, overexpressing CaN in young animals induces aging-like deficits of LTP, and deactivating CaN increases the synaptic strength in aged animals, which facilitates LTP [218]. Similarly, Ca^2+^-dependent CaN activation results in LTD by removing NMDAR and AMPAR via endocytosis in aged or APP Tg mice [180,219]. By inhibiting the activity of CaN, LTP is induced by inhibitors or Aβ in APP and Tg2576 mice [28,180].

With respect to Ca^2+^ transporters in the cell membrane, Aβ oligomers induce the dysfunction of Ca^2+^ and inhibit LTP in an NMDAR-dependent mechanism [220]. In addition, NMDAR mediated the entry of Ca^2+^ into spines and dendrites, which resulted in insufficient activation of LTP in the rat hippocampus [221]. Interestingly, NMDAR-dependent LTD requires transient incorporation of Ca^2+^-permeable (CP)-AMPAR into the synapse, which is mediated by AKAP150-anchored PKA and calcineurin [222]. Consistent with this observation, infusion of Aβ oligomers into the CA1 region of the hippocampus resulted in a rapid insertion of CP-AMPAR into synapses [165]. More directly, AMPAR mediated the effects of Ca^2+^, increasing not only LTP but also LTD. The mutation of GluR2, a subunit of AMPAR, obviously induced LTP in hippocampal slices [223]. CP-AMPAR insertion into synapses was required for the induction of LTP, which was induced by specific stimuli, leading to the assembly of heteromeric AMPARs containing both GluA1 and GluA2 subunits in CA1 hippocampal neurons [224]. In addition, CP-AMPAR mediated the effects of glycine on the induction of LTP-dependent spine enlargement via CaMKI-activating mechanisms in mature hippocampal neurons [225]. In cultured rat hippocampal neurons, Ca^2+^/calmodulin binding to PSD-95 induced the loss of synaptic PSD-95 and surface AMPARs, which resulted in activated LTD [226].

In addition to NMDAR and AMPAR, the activation of VGCC induced LTP via CaMKII in hippocampal slides [227]. In addition, Cav1.2 expression is essential for LTP, synaptic plasticity, and memory in the hippocampus [228]. As Ca^2+^ transporters in the cell membrane, TRPs are involved in regulating synaptic plasticity. For example, TRPV1 activation by capsaicin and resiniferatoxin induces a switch from LTD to LTP by enhancing Ca^2+^ influx [229]. Treatment with the agonist of TRPV1 and 4-endocannabinoid anandamide (AEA) induced LTP in CB1^−/−^ or TRPV1^−/−^ mice [230,231]. In addition, the inhibition of TRPM2 enhanced LTP in traumatically injured brains of mice [232]. In contrast, TRPM4 reduction eliminated NMDAR-dependent LTP in CA1 hippocampal neurons [233].

## 17. ER Transporters Are Responsible for Releasing Ca^2+^ from Internal Stores, Leading to Regulate the Synaptic Plasticity

With respect to intracellular Ca^2+^, LTD is induced via InsP3-mediated Ca^2+^ influx mechanisms [234]. Similarly, the activation of metabotropic glutamatergic receptors induced the production of InsP3 to release Ca^2+^ from internal stores, which resulted in promoting LTD in hippocampal slices [235]. Blocking InsP3R led to a switch of LTD to LTP and the elimination of heterosynaptic LTD, whereas blocking RyR eliminated both LTP and homosynaptic LTD at synapses that were activated, normally at low frequencies, in rat hippocampal slides and 3 × Tg mice [196,236]. In addition, knocking out the expression of RyR3 concurrently increases LTP and reduced LTD [237,238]. As critical genes for AD, presynaptic inactivation of PSs impairs LTP by controlling RyR-mediated Ca^2+^ release from the ER [239]. As Ca^2+^ depletion from the ER induces SOCE, it is reasonable to speculate that SOCE is involved in regulating synaptic plasticity. In FVB/NJ mice, reduction of SOCE-mediated Ca^2+^ entry reduced CaMKII activity, leading to destabilization of the mushroom spine and reducing LTP-mediated memory formation [240]. In the same experimental model, the overexpression of STIM1 in mouse brain neurons enhanced contextual learning and attenuated long-term depression [240]. With respect to mitochondrial Ca^2+^ stores, knocking out the expression of VDAC1 disrupts synaptic plasticity [159]. Similar to the effects of inhibiting mPTP by cyclosporine A, porin-deficient mice showed deficits in long- and short-term synaptic plasticity [241]. Based on these observations, these transporters mediated the regulatory effects of Ca^2+^ on synaptic plasticity (Table 4).

## 18. Conclusions

During the development and progression of AD, Ca^2+^ is elevated in the cytosol of neuronal cells via its transportation from the extracellular space and intracellular stores through transporter-dependent mechanisms. Ca^2+^ accumulated in neuronal cells has the ability to induce the production and deposition of Aβ and hyperphosphorylated tau in APs and NFTs, leading to impaired learning ability in AD patients. Moreover, transporters in the cell membrane, endoplasmic reticulum, mitochondria and lysosomal membranes are critical for mediating the effects of Ca^2+^ on synaptic plasticity, which contribute to the cognitive decline associated with AD.

## Figures and Tables

**Figure 1 ijms-22-05900-f001:**
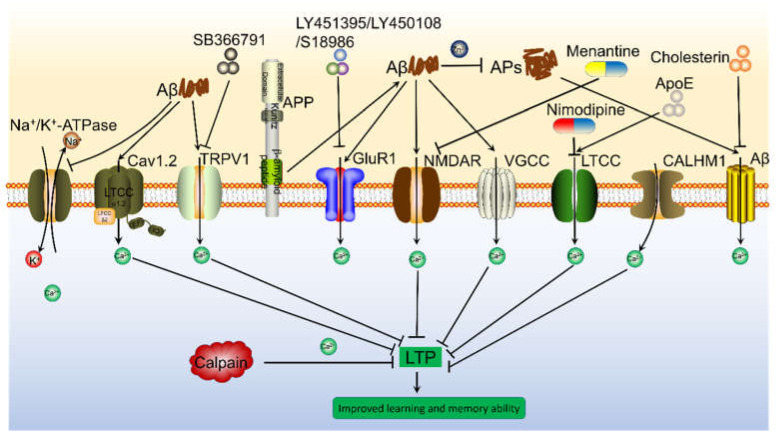
Aβ is involved in regulating Ca^2+^ influx via modulating Ca^2+^ transporters on the neuronal membranes, which result in depressing LTP and inducing cognitive decline of AD animals. Aβ can activate Ca^2+^ transporters, including NMDAR, AMPAR, LTCC, Na^+^/K^+^-ATPase, CALHM1, TRPV1 and Cav1.2 etc., which result in promoting Ca^2+^ entry into the cytoplasm, leading to elevate the concentration of Ca^2+^ in the neuronal cells. In addition, oligomeric Aβ can not selectively increase Ca^2+^ permeability of cell membrane, leading to the influx of Ca^2+^ from the extracellular space. More importantly, these transporters of Ca^2+^ have the ability to mediate the effects of Ca^2+^ on the synaptic plasticity via different mechanisms.

**Figure 2 ijms-22-05900-f002:**
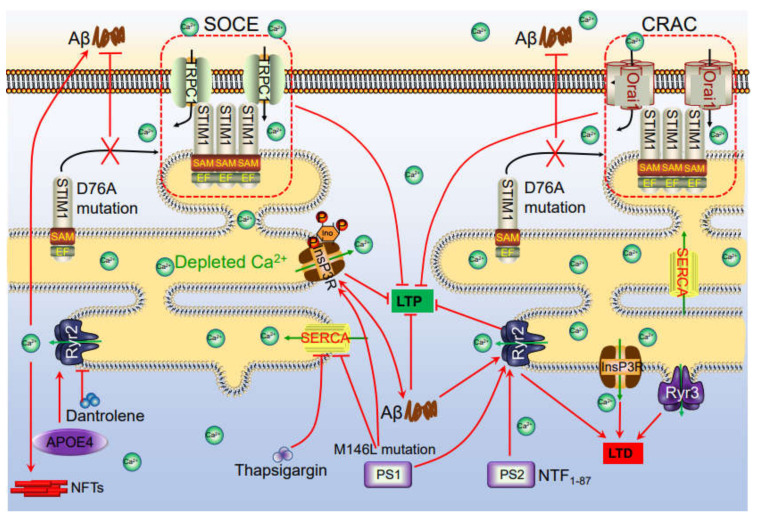
Ca^2+^ channels in the ER involved in regulating phosphorylation of tau, production of Aβ, which deposited in APs and NFTs, leading to impair learning ability via influencing synaptic plasticity. The accumulation of Aβ in the neuronal cells induces the Ca^2+^ influx from the intracellular Ca^2+^ store, ER. In addition, Ca^2+^ depletion from ER triggers a sustained extracellular Ca^2+^ influx to the cytosol via a SOCE pathway, including TRPC1 and Orai1 by activating the STIM. During these processes, InsP3R and RyR2 played important roles in inducing Ca^2+^ influx from ER to cytosol, which results in regulating synaptic plasticity, phosphorylation of tau, deposition of Aβ, leading to cognitive impairment.

**Figure 3 ijms-22-05900-f003:**
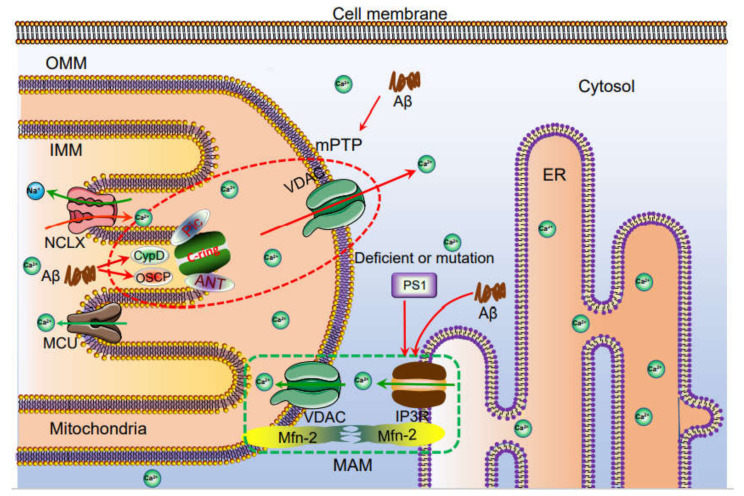
The mechanisms of Ca^2+^ transportation between mitochondria and ER. Ca^2+^ is taken up to the mitochondria via MCU. Under physiological or pathological conditions, Ca^2+^ is continuously shuffled between ER and mitochondria via VDAC. Moreover, Ca^2+^ in mitochondria induces the formation of mPTP, which traversed Ca^2+^ and small molecules, such as ROS and cytochrome C from mitochondria to cytosol, leading to the potential apoptosis of neurons. The loss of neurons will cause the cognitive dysfunction. Deficient or mutation: Defective PS1 due to exon 9 deletion (ΔE9), as well as PS1^M146V^ or PS1^L286V^ mutations, lead to Ca^2+^ flow to mitochondria via mitochondria associated endoplasmic reticulum membrane, (MAM), which further promotes apoptosis.

**Figure 4 ijms-22-05900-f004:**
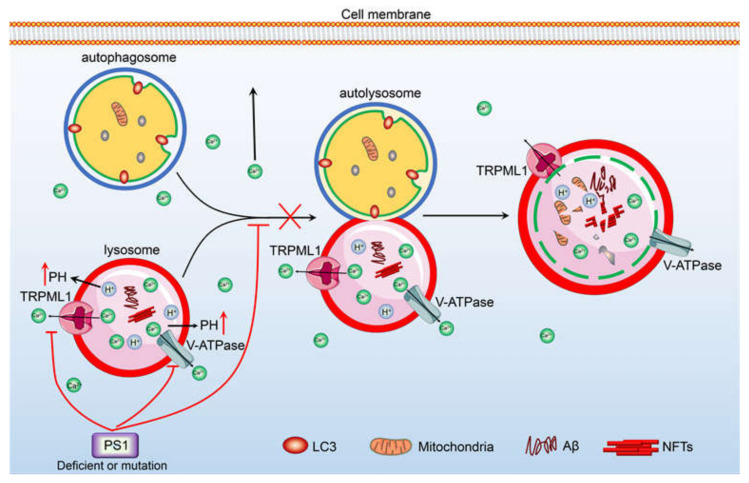
Ca^2+^ potentially contribute to regulate the degradation of Aβ and the deposition of hyperphosphorylated tau via its transporters, including v-ATPase and TRPML1 etc., in the membrane of lysosome. TRPML1 and v-ATPase are responsible for inducing the efflux of Ca^2+^ from lysosome. The accumulation of Ca^2+^ in the cytosol can stimulate the phosphorylation of tau in the neurons, leading to the deposition of hyperphosphorylated tau in NFTs. In addition, the loss of PS1 induces the release of Ca^2+^ into the cytosol via TRPML1, which results in blocking the fusion between autophagosome and lysosome, leading to prevent the degradation of Aβ.

**Table 1 ijms-22-05900-t001:** The levels of Ca^2+^ are elevated in the AD patients and animal models.

Cat.	Stimulator or Mediator	Mechanism	Experimental Model	Reference
Aβ	Aβ_1–40_	Aβ_1–40_→IL-1β→Ca^2+^ influx	Rat cortical synaptosomes and cultured cortical neurons	[29]
	Aβ_25–35_	Aβ_25–35_→L-/T-VGCC→Ca^2+^ influx	Rat CA1 pyramidal neurons	[31]
	Aβ	Aβ→Ca^2+^ influx	APP/PS1 Tg mice	[30]
		Aβ→PKA∪L-VGCC→Ca^2+^ influx	Neurons	[34]
	APs	Ca^2+^ in the spines and dendrites of cortical pyramidal neurons of APs → Ca^2+^ in the adjacent resting neurons.	The spines and dendrites of cortical pyramidal neurons in 3 × Tg AD animals	[22]
		APs→Ca^2+^ influx	The astrocytes of 6-month-old APP/PS1 mice	[35]
		Aβ→Formation of cation channels→Ca^2+^ passage	Artificial lipid membranes	[39]
		Oligomeric Aβ→Ca^2+^ influx and leakage from intracellular Ca^2^^+^ stores	SH-SY5Y cells	[35]
		Aβ→Formation of pores in the cell membrane of post-mortem→Ca^2+^ influx	Post-mortem of AD brains	[36]
	sAPP	sAPP→cGMP→K^+^ channel┤Ca^2+^	Hippocampal neurons	[56]
	γ-secretase	γ-secretase→ER-Ca^2+^	SH-SY5Y cells (control and PSEN2^T122R^-expressing)	[61]
CM	NMDAR	memantine nitrate-06 (MN-06)┤NMDAR→Ca^2+^ influx	Primary rat cerebellar granule hippocampal neurons	[64]
		Aβ∪endogenous Ca^2+^ channels→ NMDAR→Ca^2+^ influx	Mature hippocampal neurons	[63]
	AMPAR	LY451395, LY450108 and S18986┤AMPAR→Ca^2+^ influx	AD animal models	[69,70,71,72]
	P/Q-VGCC	Aβ┤P/Q-VGCC→Ca^2+^ influx	Hippocampal neurons	[73]
	N/T/L-VGCC	Aβ_1–40_→N/T/L-VGCC→postsynaptic Ca^2+^ response	Cortical neurons	[29,74,128]
	Na^+^/K^+^-ATPase	Aβ┤ion-motive ATPases┤NMDAR and VGCCs→Ca^2+^ influx Aβ┤Ca^2+^-ATPase┤Ca^2+^ efflux	Primary neurons and synaptosomes of adult post-mortem hippocampus	[76]
	CALHM1	Voltage∪extracellular Ca^2+^→CALHM1	hippocampal slices from wild-type Calhm1^+/+^, Calhm1^+/−^, and Calhm1^−/−^ mice	[78]
	APOE	APOE→G-protein-linked PLC→Ca^2+^ influx and mobilization	Neurons	[79]
		APOE4>E3>E2→P/Q type Ca^2+^-channels→ intracellular free Ca^2+^	Rat hippocampal astrocytes and neurons	[80]
		APOEε4→ intracellular Ca^2+^	Primary cultured astrocytes of APOE^−/−^ mice	[81]
ER	Aβ/InsP3R	Aβ→InsP3R→Ca^2+^ response	Cultured neurons	[87]
	Aβ_1–42_/RyR	Aβ_1–42_→RyRs→Ca^2+^ flux	primary cultured hippocampal neurons	[88]
	Aβ aggregates/InsP3R/RyR	Aβ aggregates→InsP3R and RyR→Ca^2+^ flux from ER	Human brain tissues and cells, hippocampal CA1 pyramidal neurons	[82,129]
	PS1/InsP3R/RyR/SERCA	PS1∪InsP3R, RyR and SERCA→Ca^2+^ signaling cascade	Primary rat cortical neurons	[89,90,91,93,94]
	PS/InsP3R	PS∪InsP3R→Ca^2+^ flux	Primary cortical neurons	[90]
	PS1^mut^/InsP3	PS1^mut^→PLC→InsP_3_→Ca^2+^ flux from ER	SH-SY5Y cell	[95]
	PS^mut^/RyR	PS^mut^→InsP3R and RyR→Ca^2+^ release from ER	PC12 cells, mouse neurons and lipid bilayers	[93,96,130,131]
	PS^mut^/SERCA	PS^mut^∪SERCA→Ca^2+^ influx	SH-SY5Y cells and patient-derived fibroblasts	[132]
	APOE4/RyR	APOE4→RyR→Ca^2+^ release from ER→APs and NFTs	Rat primary hippocampal neurons	[98,99,100]
	Stim1^D76A^	Stim1^D76A^ mutation┤SOCE→Ca^2+^ influx	Primary neurons from the PS1^mut^ mice	[104,105]
	Stim2	PS1^M^^146V^ mutation┤STIM2→SOCE→Ca^2+^ influx	PS1^M^^146V^ mice	[106]
	Stim1	PS1 ΔE9 mutation→Stim1→SOCE→Ca^2+^ influx	mouse hippocampal neurons	[107]
	TRPC3	BDNF→TRPC3→Ca^2+^ influx.	Pontine neurons and SH-SY5Y cells	[108,110]
	TRPC6	PS2→TRPC6┤Ca^2+^ influx	HEK293 cells	[109]
MT	PS1^L^^286V^ and PS1^M^^146L^	PS1^L^^286V^ mutation┤Mitochondria→Ca^2+^ flux	PS1^L^^286V^ mutated PC12 cells and PS1^M^^146L^ lymphoblasts	[111,112,113]
	VDAC	hAPPSwe→VDAC1→Ca^2+^ flux to the mitochondria	Tg2576 mice	[114]
	MCU	MCU→Ca^2+^ flux to the mitochondrial matrix	COS-7 cell	[115,116]
	Na^+^/Ca^2+^ exchanger	Na^+^/Ca^2+^ exchanger→Ca^2+^ across IMM	HEK293T cells	[117,118,119]
	mPTP	mPTP→Efflux of Ca^2+^ from mitochondria	SH-SY5Y cells	[120]
LM	v-ATPase/CAX	V-ATPase and CAX→Ca^2+^ influx to lysosomes	Rat kidney fibroblasts	[122,123,133]
	TRPML/TPC	TRPML and TPC→Ca^2+^ efflux from lysosomes	HEK293 cells	[124]
	VGCC	VGCC→Ca^2+^ release┤autophagic fusion and/or autophagy flux.	Cacna1a^−/−^ and Cacna2d2^−/−^ mice	[125]
	PS1^mut/−^	Mutation or deletion of PS1┤v-ATPase →Ca^2+^ uptake by lysosomes	APP/PS1 mice	[126]
	PS1/2^−/−^	PS1 and 2 knockout┤Ca^2+^ uptake by lysosomes→autophagy process	PS1/2^−/−^ neurons	[127]

**Table 2 ijms-22-05900-t002:** The roles of Ca^2+^ in the production and depostion of Aβ as well as the phosphorylation of tau.

Cat.	Stimulator or Mediator	Mechanism	Experimental Model	Reference
Ca^2+^	Aβ	Ca^2^^+^ ionophore, A23187→free Ca^2^^+^→Aβ production	hAPP overexpressed HEK293 cells, Primary cultured neurons from 3 × Tg AD mice	[134,135,136]
		Ca^2+^→Aβ	SH-SY5Y cells	[36]
		Ca^2+^→Aβ_1–40_ oligomers	Neurons	[137]
		Ca^2^^+^→Aβ fibrils	AD mice and in vitro Aβ peptides	[138,139]
CM	NMDAR	Memantine┤NMDAR→Aβ	SH-SY5Y cells	[138]
	AMPAR	AMPAR→Ca^2+^→tau phosphorylation	PS1^mut^ mice	[140,141]
		Memantine┤NMDAR→Aβ_1–40_	APP23 mice	[142]
		NMDAR→ADAM10	Primary mouse cortical neurons	[143]
	AMPAR	AMPAR→α-secretase→sAPPα┤Aβ	Cortical neurons	[144]
	CALMH1	CALHM1^P86L^→sAPPβ→Aβ	APP Tg mice	[138,145]
	L-VGCC	L-VGCC→Ca^2+^→Aβ	Rat cortical neurons	[134,138]
	Cav1.2	Isradipine┤Cav1.2→Aβ	3 × Tg mice	[34]
	APOE4	APOE4→Aβ42 in CSF	AD patients	[34]
	APOE	APOE1-3┤Aβ	hAPOE isoforms (PDAPP/TRE) expressing Aβ-amyloidosis mice	[146]
ER	InsP3R	InsP3R^−/−^ receptor┤Aβ	InsP3R^−/−^ Sf9 and DT40 cells	[90]
	RyR	RyR→NFTs	AD patients, Primary cultured rat neurons	[101,147]
		RyR→Ca^2+^→Aβ	βAPP expressed HEK293 cells	[134,135]
		Dantrolene→RyR→β-/γ-secretase→phosphorylation of APP and formation of APs	Dantrolene treated AD mice	[148,149]
	RyR2	APP mutation→RyR2^PTM^→Ca^2+^ leaky┤Aβ	SH-SY5Y cells	[150]
		FKBP12.6∪RyR2→Ca^2+^ leaky┤APs	3 × Tg mice	[150]
	RyR3	RyR3^−/−^┤APs	APP/PS1 mice	[151]
	SERCA	Thapsigargin or siRNA┤SERCA→Aβ	PS1^−/−^ and PS2^−/−^ fibroblasts	[94]
		Thapsigargin┤SERCA→Ca^2+^→Aβ	APP overexpressed HEK293 cells	[135]
		10 nM thapsigargin→Aβ 20 nM thapsigargin┤Aβ	APP overexpressed CHO cells	[152]
	Stim1/Orail	Stim1/Orai1→SOCE→Ca^2+^→Aβ/APs	APP expressed HEK293 cells	[105]
	SOCE	SOCE→mushroom spines ┤Aβ┤memory functions	PS1^M^^146V^ knockin hippocampal neurons	[153,154]
		SOCE→Ca^2+^ influx ┤Aβ→AD	Human neuroblastoma cells, Primary cultured hippocampal neurons	[155,156]
		SOCE inhibition→Aβ_1–42_	SH-SY5Y cells, Human neuroglioma H4 cells	[157,158]
MT	VDAC1	Reduced expression of VDAC1┤βAPP, Tau, PS1, PS2, and BACE1	VDAC1^+/−^ vs VDAC1^+/+^ mice	[159]
	mPTP	APP^KM670/671NL^/PS1^L^^166P^∪dutasteride┤mPTP→APs	Primary neurons and APP/PS1 Tg mice	[160]
Ca^2+^	p-tau	Ca^2+^→p-tau	SH-SY5Y cells	[100]
		Ca^2+^→GSK3β→p-tau	SH-SY5Y cells	[161]
		Ca^2+^→p-tau	Primary hippocampal neurons and the immortalized GnRH neurons (GT1-7 cells).	[162]
		Ca^2+^→mPGES-1/PGE_2_/EPs/CDK5/p35/p25→p-tau	N2a and APP/PS1 Tg mice	[19]
	NFTs	Ca^2+^→Ca^2^^+^-activated kinases→p-tau→NFTs	SH-SY5Y, N2a and AD mice models	[100,163]

**Table 3 ijms-22-05900-t003:** Ca^2+^ accelerates the cognitive decline of AD.

Cat.	Stimulator or Mediator	Mechanism	Experimental Model	Reference
Ca^2+^		Serum Ca^2+^→cognitive decline	Aging people	[174]
		Ca^2+^→dementia	AD patients	[175]
	Aβ oligomes	Aβ oligomers→Ca^2+^ influx┤LTP→synaptic plasticity→learning and memory	AD models, Hippocampal slices and APP/PS1 Tg mice	[176,177,178]
	Calpain	Inhibitor┤calpain→Aβ┤learning and memory	APP/PS1 mice	[179]
	Calcineurin	Inhibitor┤calcineurin┤learning and memory	Tg2576 mice	[180]
CM	NMDAR	Calcineurin→removing NMDAR/AMPAR by endocytosis┤cognition of AD	APP/PS1 mice	[181]
		Antagonist┤NMDAR┤synaptic plasticity┤cognitive decline	Rats	[182,183]
		Blocking NMDAR┤Ca^2+^┤cognition	AD patients and AD mouse models	[184,185]
		CP-AMPAR→Ca^2+^ influx→neuronal network dysfunction/excitotoxicity→cognitive decline	APP/PS1 mice	[186]
	L-VGCC	L-VGCC→Ca^2+^ currents→cognitive decline	CA1 synapses of 3 × Tg AD mice	[187]
		Nifedipine┤Ca^2+^ channel→cognitive impairment	KK-A(y) mice	[188]
		Nimodipine┤L-VGCC┤learning ability	Mild-to-moderate AD patients	[189]
	T-VGCC	ST101┤T-VGCC┤LTP/p-CaMKII →cognitive decline	Rat cortical slices	[190]
	NMDAR	MK-801┤NMDAR→Ca^2+^→cognitive decline	Traumatic brain injury (TBI) mice	[191]
	Cav 2.1	Cav 2.1^−/−^┤Ca^2+^┤learninig ability	Cav 2.1 knocking out mice	[192]
	TRPV1	SB366791┤TRPV1┤cognitive performance	Dopamine D3 receptor (D3R)^−/−^ mice	[193]
	APOE4	APOE4→serum Ca^2+^┤cognitive function	Aging people	[194]
	CALHM1	CALHM1^P86L^ polymorphism→AD	Chinese populations	[195]
ER	InsP3	PS1^M^^146V^┤InsP3→InsP3R1→Ca^2+^ →memory loss	PS1^M^^146V^ mice	[196]
	InsP3R	SOCE∪InsP3R→Ca^2+^┤cognitive impairment	Sporadic or mild AD patients	[197]
	RyR	Dantrolene┤RyR┤synaptic plasticity→cognitive ability	AD mouse model	[198]
	RyR2/RyR3	RyR3^−/−^/RyR2^+/+^┤social behavior and memory	RyR3^−/−^/RyR2^+/+^ mice	[199,200]
		RyR^PTM^→ER→Ca^2+^ leaky →cognitive deficits	3 × Tg mice	[150]
	Stim2/SOCE	STIM2^−^∪SOCE^−^┤mushroom spines→LTP→memory	PS^mut^ mice	[106,201]
		SOCE^−^→cognitive decline→AD	Hippocampal slice cultures	[202]
MT	VDAC1	VDAC1∪p-tau, Aβ, and γ-secretase→neurotoxicity→cell death→dementia→AD	APP, APP/PS1 and 3 × Tg mice	[203]
	mPTP	DS16570511, DS44170716┤MCU→Ca^2+^ influx to mitochondria→mPTP→apoptotic cell death	HEK293 cells	[204,205]
LM	TPC	Tetrandrine, NED-19┤TPCE2┤re-acidify lysosome→autophagy	MEFs cells	[206]
		Beclin1^−/−^→Aβ	hAPP mice	[207]

**Table 4 ijms-22-05900-t004:** The roles of Ca^2+^ in synaptic plasticity.

Cat.	Stimulator or Mediator	Mechanism	Experimental Model	Reference
Ca^2+^		Aβ→Ca^2+^ influx→LTD┤memory┤AD	Tg2576 mice	[28]
		Aβ oligomers→Ca^2+^┤LTP	Hippocampal slices	[176]
		PS1^−/−^┤LTP	PS1^−/−^ mice	[216]
		BAPTA-AM┤Ca^2+^┤LTP.	Aged rat hippocampal slices	[217]
CM	CaN	CaN^+^┤LTP CaN^−^→synpatic strength →LTP	CaN^+^ mice	[218]
		Ca^2+^→CaN→LTD	Aged or APP mice	[180,219]
		Inhibitors┤CaN┤LTP	APP mice	[180]
		Aβ┤CaN┤synaptic plasticity	Tg2576 mice	[28]
	NMDAR	Aβ oligomers→NMDAR→Ca^2+^┤LTP	Hippocampal CA1 and DG regions	[220]
		NMDAR→Ca^2+^┤LTP	Rat hippocampus	[221]
	AMPAR	AMPAR→Ca^2+^→LTP∪LTD	CA1 pyramidal cells	[242]
		GluR2^−/−^→LTP	GluR2^−/−^ mice	[223]
		CP-AMPAR→LTP	CA1 hippocampal neurons	[224]
		Glycine→CP-AMPAR→CaMKI→LTP	Mature hippocampal neurons	[225]
		Ca^2+^/Calmodulin∪PSD-95┤PSD-95∪AMPAR┤LTD	Rat hippocampal neurons	[226]
	VGCC	VGCC→CaMKII→LTP	Hippocampus slides	[227]
	Cav1.2	Cav1.2^+^→LTP, synaptic plasticity, and the memory	Ca(V)1.2 (cKO) mice	[228]
	TRPV1	Capsaicin and resiniferatoxin→TRPV1→LTP	Hippocampus slides	[229]
		Capsaicin→TRPV1→Ca^2+^ influx→LTP	Hippocampus slides	[229]
	TRPV1/4	Endocannabinoid anandamide (AEA) →TRPV1/4→LTP	CB1^−/−^ mice TRPV1^−/−^ mice	[230,231]
	TRPM2	Inhibitor┤TRPM2┤LTP	Traumatic injured brain of mice	[232]
	TRPM4	TRPM4^−^┤NMDAR→LTP	CA1 hippocampal neurons	[233]
ER	IP3	IP3→Ca^2+^ efflux from ER→LTD	Myosin-Va mutation mice or rats	[234]
		Metabotropic glutamatergic receptors→InsP3→Ca^2+^ efflux from ER→LTD	Hippocampal slices	[235]
	InsP3R/RyR	InsP3R^−^→LTP∪┤LTD RyR^−^┤LTP∪LTD	Rat hippocampus slides, 3 × Tg AD mice	[196,243]
	RyR	RyR3^−/−^→LTP∪┤LTD	RyR3^−/−^ mice, 3 × Tg mice	[237,238]
		PS^−^→RyR→Ca^2+^ release from ER ┤LTP	PS conditioned neurons from CA1 and CA3	[239]
	SOCE	SOCE^−^┤Ca^2+^ influx→CaMKII→LTP→memory	FVB/NJ mice	[240]
		STIM1^+^┤LTD┤contextual learning	FVB/NJ mice	[240]
MT	VDAC	VDAC1^−/−^┤synaptic plasticity	VDAC1^−/−^ mice	[159]
	mPTP	Cyclosporine A┤mPTP→long and short term synaptic plasticity	Porin-deficient or cyclosporin A-treated mice	[241]

CM, cell membrane; MT, mitochondria; LM, lysosome; PTM, post-translational modification; →, stimulate, activate, induce, result in, lead to; ┤, inhibit, block, suppress, deactivate, degrade; +, overexpress, activate, upregulate, induce; −, knockdown, deplete, ablate, siRNA, deactivate, downregulate, deficiency; −/−, knock out; ∪, interact, facilitate, associate, potentiate, recruit, and.

## Data Availability

Not applicable.

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
