# Peer review of "Elevating the Levels of Calcium Ions Exacerbate Alzheimer’s Disease via Inducing the Production and Aggregation of β-Amyloid Protein and Phosphorylated Tau"

_ijms, 2021, doi:10.3390/ijms22115900_

Round 1

Reviewer 1 Report

This review mainly explores the molecular mechanisms by which a Ca2+ imbalance in AD affects the regulation of Aβ, tau, and neural plasticity, specifically from the perspective of Ca2+ transporters in cell, mitochondrial, endoplasmic reticulum (ER) and lysosomal membranes. Authors have methodically discussed the role of Ca2+ influx through membrane channels, intracellular Ca2+ reservoirs (ER,  Lysosome, mitochondria) and their impact on Ca2+ homeostasis as well as in lysosomal degradation of intracellular proteins. 

The review is well discussed with most possible connection of Ca2+ with Aβ, tau in Alzheimer's disease. Figures are also very informative and clear.

The author need to put some more references in introduction where they have mentioned that Ca2+ metabolic disorder was evident before AP or NFT formation.  As this is the main leading statement of this whole review and indicates that, the alteration Ca2+ homeostasis is earlier event in the development of AD, which makes the ca2+ pathways very important for study and therapeutic target.

Author Response

Response to Reviewer 1 Comments

Author’s response

According to the reviewer’s comment, we have carefully checked and revised the spelling of the English language and style throughout the manuscript (Page 3, para 2, line 120; Page 4, para 1, line 155; Page 4, para 3, line 182-183; Page 6, para 1, line 225 and 230; Page 6, para 2, line 233, 235 and 242-243; Page 7, para 2, line 271, 277 and 296-297; Page 9, para 2, line 336; Page 11, para 1, line 398 and 401; Page 11, para 3, line 423-424; Page 14, para 1, line 544).

This review mainly explores the molecular mechanisms by which a Ca2+ imbalance in AD affects the regulation of Aβ, tau, and neural plasticity, specifically from the perspective of Ca2+ transporters in cell, mitochondrial, endoplasmic reticulum (ER) and lysosomal membranes. Authors have methodically discussed the role of Ca2+ influx through membrane channels, intracellular Ca2+ reservoirs (ER, Lysosome, mitochondria) and their impact on Ca2+ homeostasis as well as in lysosomal degradation of intracellular proteins.

Author’s response

Thanks!

Reviewer’s Comment #1:

The review is well discussed with most possible connection of Ca2+ with Aβ, tau in Alzheimer's disease. Figures are also very informative and clear.

Author’s response

Thanks for the reviewer’s comments and insightful suggestions for improving the quality of our manuscript.

Reviewer’s Comment #2:

The author needs to put some more references in introduction where they have mentioned that Ca2+ metabolic disorder was evident before AP or NFT formation. As this is the main leading statement of this whole review and indicates that, the alteration Ca2+ homeostasis is earlier event in the development of AD, which makes the Ca2+ pathways very important for study and therapeutic target.

Author’s response

According to the reviewer’s comments, we provide more references to show that Ca2+ metabolic disorder was evident before AP or NFT formation (Yu, et al., 2009; Cao, et al., 2019a; b).

Reviewer 2 Report

Review of a manuscript “Elevating the levels of calcium ions exacerbate Alzheimer’s disease via inducing the production and aggregation of β-amyloid protein and phosphorylated tau” by Pei-Pei Guan and coauthors submitted to IJMS.

Alzheimer's disease (AD) is a severe highly prevalent neurodegenerative disease for which there is no efficient treatment changing the course of the disorder. Among several factors contributing to AD the imbalance of Ca2+ causes dysregulation of metabolic pathways and affects many neurophysiological functions and biochemical alterations. The authors summarized recent data and explored some molecular and cellular mechanisms by which a Ca2+ imbalance may affect the regulation of Aβ, tau, and neural plasticity. They also discussed a role of Ca2+ transporters in cell, mitochondrial, lysosomal and endoplasmic reticulum (ER) membranes. This is an important direction of biomedical science and the data presented in the manuscript will be interesting to the readers of IJMS.

The following corrections should be done:

Abstract

Lines 17-22 A: “In view of this, the current review summarizes the mechanisms by which Ca2+ is transported into and out of cells and organelles to affect the balance of intracellular Ca2+ levels and Ca2+ metabolism and explores the important roles of these transport mechanisms in the pathogenesis of AD”.

B: “Specifically, we mainly focus on the molecular mechanisms by which Ca2+ is transported through the cell, endoplasmic reticulum, mitochondrial and lysosomal membranes”

There are redundant statements in these two sentences (A and B) shown by Italics.

Abstract should be concise and succinct, so the authors should rewrite this part avoiding the repetitions.

Introduction:

Lines 30-31: After “Alzheimer's disease (AD), commonly known as dementia, is a neurodegenerative disease with a high incidence rate” the authors should add the following sentence and a reference:

AD may share common biological pathways and is often associated with diabetes and other comorbidities (ref. Caveolin: A New Link Between Diabetes and AD. Cell Mol Neurobiol. 2020; 40 (7):1059-1066).

Lines 84-85 :” According to the pathogenesis of AD,…” This fragment does not bring any sense and should be deleted.

Lines 93-94:”Because of the self-aggregating characteristics of Aβ, the concentration of Ca2+ in the spines and dendrites of cortical pyramidal neurons around APs was higher than the normal value exhibited by adjacent resting neurons (Tong et al., 2018).”

The sentence should be rewritten as follows:” Because of the self-aggregating characteristics of Aβ, the concentration of Ca2+ in the spines and dendrites of cortical pyramidal neurons around APs is higher than the normal value in adjacent resting neurons (Tong et al., 2018).”

Lines 93-135: This is a very long fragment which is hard to read. It should be either truncated or split into several fragments separated by indented lines/periods.   

Figure 3

It is unclear to what exactly “Deficient or mutation” is related. Should be explained in the legend.

References

Line 577, line 770: Some references are obsolete and should be replaced by more recent ones, for example,  (Obenaus, et al., 1989);  Fox, et al., (1987).

Author Response

Response to Reviewer 2 Comments

Review of a manuscript “Elevating the levels of calcium ions exacerbate Alzheimer’s disease via inducing the production and aggregation of β-amyloid protein and phosphorylated tau” by Pei-Pei Guan and coauthors submitted to IJMS.

Alzheimer's disease (AD) is a severe highly prevalent neurodegenerative disease for which there is no efficient treatment changing the course of the disorder. Among several factors contributing to AD the imbalance of Ca2+ causes dysregulation of metabolic pathways and affects many neurophysiological functions and biochemical alterations. The authors summarized recent data and explored some molecular and cellular mechanisms by which a Ca2+ imbalance may affect the regulation of Aβ, tau, and neural plasticity. They also discussed a role of Ca2+ transporters in cell, mitochondrial, lysosomal and endoplasmic reticulum (ER) membranes. This is an important direction of biomedical science and the data presented in the manuscript will be interesting to the readers of IJMS.

Author’s response

Thanks!

The following corrections should be done:

Reviewer’s Comment #1:

Abstract

Lines 17-22 A: “In view of this, the current review summarizes the mechanisms by which Ca2+ is transported into and out of cells and organelles to affect the balance of intracellular Ca2+ levels and Ca2+ metabolism and explores the important roles of these transport mechanisms in the pathogenesis of AD”.

B: “Specifically, we mainly focus on the molecular mechanisms by which Ca2+ is transported through the cell, endoplasmic reticulum, mitochondrial and lysosomal membranes”

There are redundant statements in these two sentences (A and B) shown by Italics.

Abstract should be concise and succinct, so the authors should rewrite this part avoiding the repetitions.

Author’s response

Following the reviewer’s comments, we had rephrased the related sentences to avoid the repetition (Page 1, para 1, line 18-23).

Reviewer’s Comment #2:

Introduction:

Lines 30-31: After “Alzheimer's disease (AD), commonly known as dementia, is a neurodegenerative disease with a high incidence rate” the authors should add the following sentence and a reference:

AD may share common biological pathways and is often associated with diabetes and other comorbidities (ref. Caveolin: A New Link Between Diabetes and AD. Cell Mol Neurobiol. 2020; 40 (7):1059-1066).

Author’s response

According to the reviewer’s comment, we have added the sentence and corresponding reference (Page 1, para 2, line 33-34).

Reviewer’s Comment #3:

Lines 84-85 :” According to the pathogenesis of AD,…” This fragment does not bring any sense and should be deleted.

Author’s response

According to the reviewer’s comment, we have deleted the phrase in the Introduction section of our revised manuscript (Page 2, para 4, line 90).

Reviewer’s Comment #4:

Lines 93-94: “Because of the self-aggregating characteristics of Aβ, the concentration of Ca2+ in the spines and dendrites of cortical pyramidal neurons around APs was higher than the normal value exhibited by adjacent resting neurons (Tong et al., 2018).”

The sentence should be rewritten as follows:” Because of the self-aggregating characteristics of Aβ, the concentration of Ca2+ in the spines and dendrites of cortical pyramidal neurons around APs is higher than the normal value in adjacent resting neurons (Tong et al., 2018).”

Author’s response

According to the reviewer’s comment, we have replaced the correct sentence in our revised manuscript (Page 3, para 2, line 99-103).

Reviewer’s Comment #5:

Lines 93-135: This is a very long fragment which is hard to read. It should be either truncated or split into several fragments separated by indented lines/periods.

Author’s response

We are sorry that we could not catch the accurate meanings of the reviewer since it is not a sentence from line 93 to 135. It spans about 42 lines and we find that it is not a sentence without truncation. If the reviewer still think that some sentences should be split, please feel free to contact us.

Reviewer’s Comment #5:

Figure 3

It is unclear to what exactly “Deficient or mutation” is related. Should be explained in the legend.

Author’s response

According to the reviewer’s comment, we have explained the “Deficient and mutation” in the Figure legends section of our revised manuscript (Page 8, para 1, line 305-307).

Reviewer’s Comment #6:

References

Line 577, line 770: Some references are obsolete and should be replaced by more recent ones, for example (Obenaus, et al., 1989); Fox, et al., (1987).

Author’s response

According to the reviewer’s comment, we have replaced these two references mentioned above with more recently published references (Page 5, para 1, line 193; page 14, para4, line 588).
